# Exploring causality between bone mineral density and frailty: A bidirectional Mendelian randomization study

**Jue-xin Shen, Yi Lu**  **\*, Wei Meng, Lei Yu, Jun-kai Wang**

Department of Orthopedics, Chongming Branch, Shanghai Tenth People's Hospital, Tongji University School of Medicine, Shanghai, China

* lyyi61today@163.com

**Data Availability Statement:** All relevant data are within the manuscript and its Supporting Information files.

**Funding:** The author(s) received no specific funding for this work.

## Abstract

### Objective

The bidirectional correlation between low bone mineral density (BMD) and frailty, despite its extensive documentation, still lacks a conclusive understanding. The objective of this Mendelian randomization (MR) study is to investigate the bidirectional causal relationship between BMD and frailty.

### Methods

We utilized summary statistics data for BMD at different skeletal sites—including heel BMD (e-BMD, N = 40,613), forearm BMD (FA-BMD, N = 8,143), femoral neck BMD (FN-BMD, N = 32,735), and lumbar spine BMD (LS-BMD, N = 28,489), alongside frailty index (FI, N = 175,226) data in participants of European ancestry. MR analysis in our study was conducted using well-established analytical methods, including inverse variance weighted (IVW), weighted median (WM), and MR-Egger approaches.

### Results

We observed negative causal estimates between genetically predicted e-BMD (IVW $\beta$ = -0.020, 95% confidence interval (CI) = - 0.038, - 0.002, $P$ = 0.029) and FA-BMD (IVW $\beta$ = -0.035, 95% CI = -0.066, -0.004, $P$ = 0.028) with FI. However, the results did not reach statistical significance after applying the Bonferroni correction, with a significance threshold set at $P$ < 0.0125 (0.05/4). There was no causal effect of FN-BMD (IVW $\beta$ = - 0.024, 95% CI = -0.052, 0.004, $P$ = 0.088) and LS-BMD (IVW $\beta$ = - 0.005, 95% CI = -0.034, 0.024, $P$ = 0.749) on FI. In the reverse Mendelian randomization (MR) analysis, we observed no causal effect of FI on BMD at various skeletal sites.

### Conclusion

Our study provides support for the hypothesis that low BMD may be a potential causal risk factor for frailty, but further research is needed to confirm this relationship. However, our findings did not confirm reverse causality.

**Competing interests:** The authors have declared that no competing interests exist.

## Introduction

Frailty is a multifaceted clinical syndrome. An international consensus group defined frailty as "A medical syndrome with multiple causes and contributors that is characterized by diminished strength, endurance, and reduced physiologic function that increases an individual's vulnerability for developing increased dependency and/or death [1]." With the aging of the population, the prevalence of frailty is escalating and is closely linked to a heightened risk of numerous unfavorable outcomes and amplified healthcare expenditures. Hence, the imperative to address frailty has emerged as an urgent public health concern [2].

The prevalence of primary osteoporosis increases with age and differs by race/ethnicity [3]. It is characterized by reduced bone mineral density (BMD) and heightened susceptibility to fractures [4]. BMD is a highly genetic trait [5] and is the gold standard used to diagnose osteoporosis. The International Society for Clinical Densitometry (ISCD) recommends measuring BMD of the lumbar spine, femoral neck, and total hip to diagnose osteoporosis in postmenopausal women and men aged 50 years or older [6].

In recent times, there has been a growing acceptance of the association between frailty and osteoporosis in the elderly population [7]. Improving BMD and treating osteoporosis can effectively intervene and manage frailty. Several previous observational studies have identified a negative association between BMD and frailty in older adults [8, 9]. However, recent reports have indicated a weak correlation between frailty and osteoporosis [10]. Furthermore, there have been studies that assess frailty as a predictor of osteoporosis [11]. These studies are susceptible to confounding factors and reverse causality [12, 13], so the causal relationship between osteoporosis and frailty remains elusive. The measure of frailty used in this study is the frailty index (FI). The frailty index is a multidimensional tool that assesses an individual's overall vulnerability and resilience. It is calculated by counting the number of health deficits or impairments present in an individual, such as chronic diseases, functional limitations, cognitive decline, and social vulnerability [14]. The FI provides a quantitative measure of frailty, allowing researchers to examine its association with various health outcomes and potential causal relationships with other factors, such as BMD in this study.

Well-conducted randomized controlled trials (RCTs) are considered the gold standard for establishing causality. However, in cases where RCTs are expensive, challenging to carry out, or ethically unfeasible, Mendelian randomization (MR) offers an alternative approach for estimating the long-term causal impact of an exposure on an outcome. MR utilizes genetic variations as instrumental variables (IVs) [15, 16]. The utilization of Mendelian randomization (MR) helps alleviate the issues related to unmeasured confounding and reverse causation. This is because genetic variants, used as instrumental variables (IVs), are randomly assigned at conception and remain unaffected as the disease progresses [12, 17].

In this two-sample MR analysis, we aimed to explore whether there is a bidirectional causal relationship between BMD and frailty.

## Material and methods

### Study design

This bidirectional MR study design is shown in (**Fig 1**). Our study initially estimated the causal influence of BMD at various locations on frailty, followed by evaluating the causal impact of frailty on BMD. To consider genetic variants as IVs, usually single nucleotide polymorphisms (SNPs), the following three assumptions must be satisfied [12, 18]. Firstly, the genetic variants must exhibit a strong correlation with the exposure of interest. Secondly, the genetic variants should be unrelated to potential confounding factors, such as gender, age, smoking, and

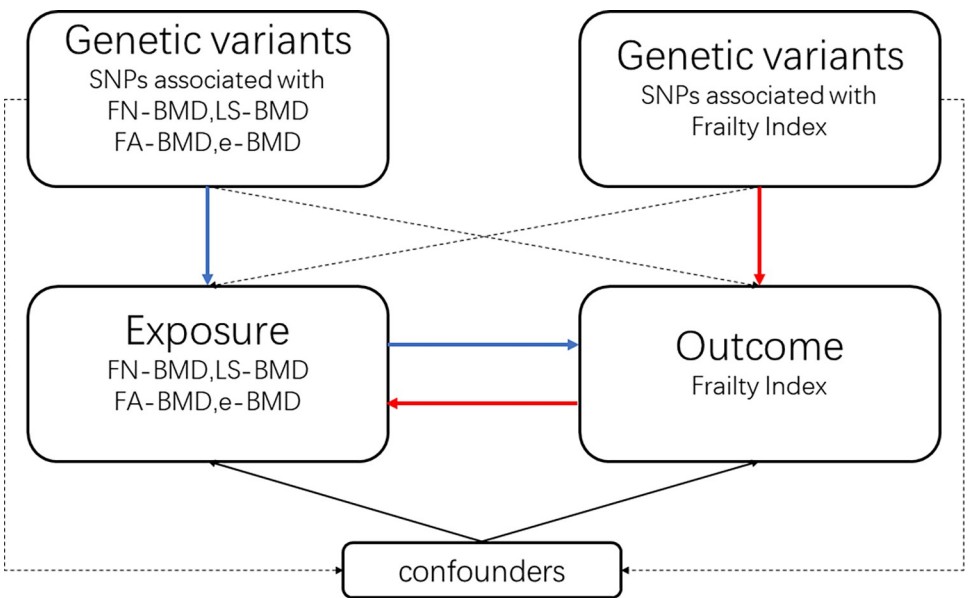

**Fig 1. In the bidirectional Mendelian randomization (MR) study design, dashed arrows represent the assumption that genetic variants are neither associated with confounders nor directly influencing the outcome, and such paths should ideally be absent in an MR analysis.** The blue and red paths illustrate the bidirectional nature of two-sample MR. SNP: single nucleotide polymorphism; BMD: bone mineral density; FN: femoral neck; LS: lumbar spine; FA: forearm; e: heel.

cardiovascular disease. Lastly, the genetic variants should not directly influence the outcome but exert their effect solely through the exposure pathway. We utilized summary statistics from genome-wide association studies (GWASs) on BMD (different skeletal sites) and frailty.

## GWAS summary data sources

Summary statistics of frailty as measured by the frailty index (FI) phenotype were derived from a recent meta-analysis of genome-wide association studies (GWAS) from the UK Biobank (UKB, N = 164,610) and TwinGene, Sweden (N = 10,616), which included 175,226 participants of European ancestry [14].

The BMD dataset consists of BMD phenotypes measured at four different skeletal sites: the femoral neck (FN), lumbar spine (LS), forearm (FA), and heel (e). Summary statistics of FN-BMD (N = 32,735), LS-BMD (N = 28,489), and FA-BMD (N = 8,143) were obtained from the Genetic Factors in Osteoporosis Consortium [19] (GeFOS) GWAS meta-analysis (53,236 participants, European origin). The e-BMD dataset (N = 40,613) was obtained from the MRC Integrative Epidemiology Unit (MRC-IEU) consortium and is based on GWAS summary statistics from the UKB. All datasets information are tabulated in the **S1 File**, and all datasets can also be accessed in the IEU Open GWAS project (https://gwas.mrcieu.ac.uk/).The e-BMD dataset and most of the FI datasets are derived from the UK Biobank, so there will be an overlap of samples [20]. Unfortunately, we were unable to locate an independent dataset for the TwinGene group, and therefore false-positive results may occur (https://sb452.shinyapps.io/overlap/).

## SNP extraction and validation

In the first step of our Mendelian randomization analysis, we applied a significance threshold ($p < 5 \times 10^{-8}$) to identify genetic instruments that were strongly associated with the exposure

in the GWAS summary data. Subsequently, we performed linkage disequilibrium (LD) clumping using the "extract_instruments" function within the TwoSampleMR package, setting an $R^2$ threshold of greater than 0.001 ($R^2 > 0.001$) and considering a clumping window of 10,000 kilobases (kb). This function facilitated the selection of independent SNPs as instruments for our exposure, based on combined GWAS significance and LD structure.

Second, summary statistics for SNPs associated with exposure were extracted from the outcome dataset. If the SNPs were not included in the outcome datasets, LD variants ($R^2 > 0.8$) were used as a proxy when possible [21]. Furthermore, we also treated SNPs as robust instruments by calculating F-statistics ($F > 10$). The calculation formula is $F = R^2 (N - k - 1)/k(1 - R^2)$, where $R^2$ represents the coefficient of determination between the SNP and the exposure, $N$ is the sample size, and $k$ is the number of SNPs used as instruments [22]. This ensures that the instrumental variables have sufficient strength to provide valid causal estimates.

Third, to assess whether the selected SNPs were associated with other traits at the genome-wide significance level, we applied the PhenoScanner database (http://www.phenoscanner.medschl.cam.ac.uk/) to check the SNPs [23]. This step helps identify potential pleiotropy or confounding effects associated with the instrumental variables used in our MR analysis.

## Mendelian randomization analysis

Prior to MR analysis, we harmonized the exposure and outcome data for SNPs in order to ensure that the $\beta$ values were assigned to the same alleles [24]. We used the inverse-variance-weighted (IVW) approach as our primary MR analysis method, which estimates the causal effect of gene-predicted exposure on outcome through a weighted regression of SNP-specific Wald ratios (i.e., $\beta$ outcome/$\beta$ exposure) [25]. To test for horizontal pleiotropy, we also performed two analyses including the MR-PRESSO global test [26] and MR Egger intercept [27]. The WM approach selects median estimates to calculate causal effects and provides consistent estimates even when up to 50% of the information comes from invalid instrumental variables [28]. We assessed the potential presence of horizontal pleiotropy using MR-Egger regression based on intercept terms, where deviation from zero ($P < 0.05$) was considered as evidence for the presence of directional pleiotropy bias [27]. To further test the robustness of results, we examined evidence of heterogeneity (a potential indicator of pleiotropy) in the IVW estimators using Cochran's Q statistic.

The results were expressed as $\beta$ and 95% confidence intervals (CIs), which provide an estimate of the change in outcome caused by each standard deviation increase in the risk factor. In our study, we performed MR analysis of BMD as exposure and FI as outcome for each of the four different skeletal sites. In view of the multiple tests, the main results were statistically significant at $P$ values $< 0.0125$ (0.05/4) after Bonferroni correction. All statistical analyses were performed bilaterally, with the help of R software (version 4.2.1) through the "TwoSampleMR" and "MRPRESSO" packages [29]. R code can be found in the **S1 File**.

## Results

### Causal effects of bone mineral density at different sites on frailty index

In our study, we extracted 53 SNPs for e-BMD, 21 SNPs for FN-BMD, 3 SNPs for FA-BMD, and 24 SNPs for LS-BMD, respectively. (**S2 File**). The MR results of this section were based on the SNPs screened under the genome-wide significance threshold ($p < 5 \times 10^{-8}$). These SNPs were derived from GWAS meta-analysis data, excluding linkage disequilibrium (LD), and p-values $< 5 \times 10^{-8}$ corresponded to an F-statistic $>30$ for every single variant. Thus, the selected IVs are robust, and "weak IVs" are negligible.

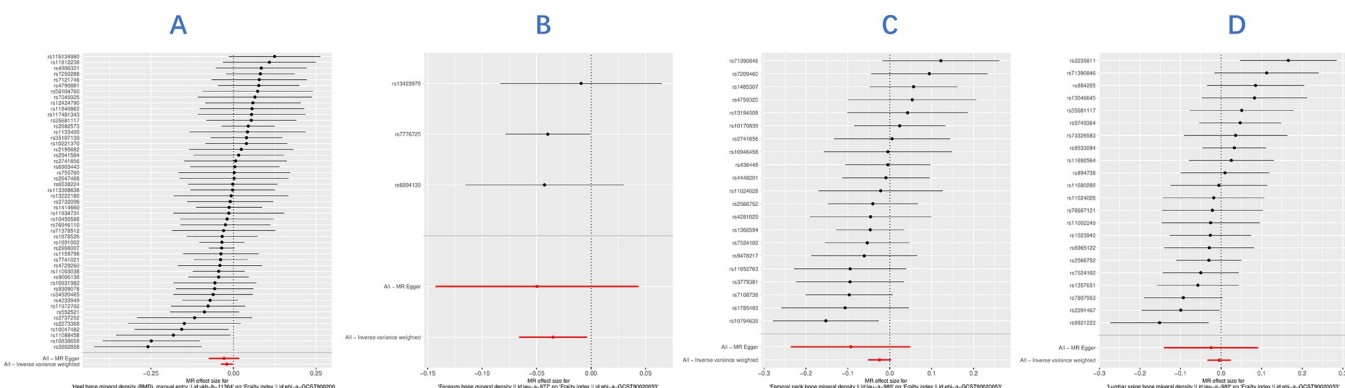

**Fig 2.** Forest plot of the causal effects of bone mineral density (BMD) associated SNPs on frailty index (FI) Figure showed the Mendelian randomization estimated effects sizes for (A) heel (e) BMD, (B) forearm (FA) BMD, (C) femoral neck (FN) BMD, (D) lumbar spine (LS) BMD. Data are expressed as β values with 95% CI.

Harmonization of data was performed before e-BMD MR analysis to remove two palindromic SNPs with moderate allele frequencies (rs2929308, rs419918). Data from other sites BMD coordinated without finding palindromic SNPs. After searching the Phenoscanner database for BMD-associated SNPs, we did not find associations with traditional risk factors for FI (e.g., smoking, body mass index, cardiovascular disease) [14, 30]. Lastly, 51 SNPs for e-BMD, 21 SNPs for FN-BMD, 3 SNPs for FA-BMD, and 24 SNPs for LS-BMD were taken as IVs. A forest plot of IVs at each site BMD was shown in the (Fig 2).

We used the inverse variance weighting (IVW) approach as the main analytical method in a random effects model (Table 1). We found negative causal estimates between genetically predicted e-BMD (IVW $\beta$ = - 0.020, 95% confidence interval (CI) = - 0.038, - 0.002, $P$ = 0.029) and FA-BMD (IVW $\beta$ = -0.035, 95% CI = -0.066, -0.004, $P$ = 0.028) with FI in the IVW model. The result of WM model was consistent with IVW. The MR-Egger result was inconsistent with IVW because its confidence interval was wider, but the direction of estimated effect is the

**Table 1. MR results for the effect of bone mineral density at different sites on the frailty index.**

| Exposures | Outcome | No. of SNPs | Method | β (95%CI) | Association P-value | Heterogeneity test | |
|---|---|---|---|---|---|---|---|
| | | | | | | Cochran's Q | P-value |
| e-BMD | Frailty index | 51 | IVW | - 0.020 (- 0.038, - 0.002) | 0.029 | 64.559 | 0.081 |
| | | | WM | - 0.035 (- 0.061, - 0.009) | 0.009 | | |
| | | | MR Egger | - 0.028 (- 0.075, -0.018) | 0.236 | | |
| FA-BMD | Frailty index | 3 | IVW | - 0.035 (- 0.066, - 0.004) | 0.028 | 0.568 | 0.753 |
| | | | WM | - 0.040 (- 0.075, - 0.004) | 0.028 | | |
| | | | MR Egger | - 0.050 (- 0.143, 0.044) | 0.486 | | |
| FN-BMD | Frailty index | 21 | IVW | - 0.024 (- 0.052, 0.004) | 0.088 | 22.609 | 0.308 |
| | | | WM | - 0.035 (- 0.072, 0.002) | 0.063 | | |
| | | | MR Egger | - 0.093 (- 0.236, 0.050) | 0.216 | | |
| LS-BMD | Frailty index | 22 | IVW | - 0.005 (- 0.034, 0.024) | 0.749 | 33.869 | 0.037 |
| | | | WM | - 0.020 (- 0.053, 0.013) | 0.233 | | |
| | | | MR Egger | - 0.025 (- 0.141, 0.092) | 0.683 | | |

Abbreviations: MR, Mendelian randomization; BMD: bone mineral density; FN: femoral neck; LS: lumbar spine; FA: forearm

e: heel; SNPs, single nucleotide polymorphisms; IVW, inverse-variance weighted; CI, confidence interval; WM, weighted median.

same. Thus, the impact on the estimation of causality was not significant. However, the results did not reach statistical significance after applying Bonferroni correction (significance threshold set at P < 0.0125 (0.05/4)). FN-BMD (IVW $\beta$ = -0.024, 95% CI = -0.052, 0.004, $P$ = 0.088) and LS-BMD (IVW $\beta$ = -0.005, 95% CI = -0.034, 0.024, $P$ = 0.749) were not causally associated with FI in the IVW models. In several sensitivity analyses, Cochran's Q test did not reveal significant heterogeneity, except for LS-BMD ($P$ = 0.037). However, this did not impact the overall MR results. MR-Egger analysis did not suggest any directional pleiotropy for the IVs ($P$ for intercept = 0.705 for e-BMD, $P$ for intercept = 0.798 for FA-BMD, $P$ for intercept = 0.347 for FN-BMD, $P$ for intercept = 0.733 for LS-BMD). The MR-PRESSO global test could not be applied to FA-BMD due to the limited number of SNPs (only 3 available). Nonetheless, the combined results from the MR Egger intercept test and the MR-PRESSO global test indicated that the Mendelian randomization analysis remained robust, unaffected by any potential horizontal pleiotropy ($P$ > 0.05) (**Table 3**).

Due to the considerable sample overlap between the e-BMD and FI datasets, caution is warranted when considering the suggested negative causal relationship.

## Causal effect of frailty index on bone mineral density at different sites

Similarly, we extracted 15 SNPs for e-BMD, 14 SNPs for FN-BMD, 14 SNPs for FA-BMD, and 14 SNPs for LS-BMD, respectively. (**S2 File**). The MR results of this section were based on the SNPs screened under the genome-wide significance threshold ($p < 5 \times 10^{-8}$) (**Fig 3**). These SNPs excluded LD and were all robust (F-statistic >10). No palindrome SNPs were found when harmonizing the data. After searching for FI-related SNPs in the Phenoscanner database, we did not find associations with risk factors for BMD (e.g., serum estradiol concentrations, smoking) [31, 32].

In the random effects model, we also used inverse variance weighting (IVW) as the main analytical method. In the IVW model, no causal relationship was found between FI and BMD for different skeletal sites (**Table 2**). e-BMD (IVW $\beta$ = 0.093, 95% CI = -0.080, 0.266, $P$ = 0.293), FA-BMD (IVW $\beta$ = 0.025, 95% CI = -0.474, 0.525, $P$ = 0.921), FN-BMD (IVW $\beta$ = 0.034, 95% CI = -0.170, 0.237, $P$ = 0.746), and LS-BMD (IVW $\beta$ = 0.048, 95% CI = -0.223,

**Table 2. MR results for the effect of frailty index on the bone mineral density at different sites.**

| Exposure | Outcomes | No. of SNPs | Method | $\beta$ (95%CI) | Association P-value | Heterogeneity test | |
|---|---|---|---|---|---|---|---|
| | | | | | | Cochran's Q | P-value |
| Frailty index | e-BMD | 15 | IVW | 0.093 (- 0.080, 0.266) | 0.293 | 15.788 | 0.326 |
| | | | WM | 0.030 (- 0.206, 0.266) | 0.802 | | |
| | | | MR Egger | - 0.853 (- 1.578, - 0.128) | 0.038 | | |
| Frailty index | FA-BMD | 14 | IVW | 0.025 (- 0.474, 0.525) | 0.921 | 18.522 | 0.139 |
| | | | WM | - 0.046 (- 0.651, 0.560) | 0.883 | | |
| | | | MR Egger | 0.276 (- 2.507, 3.060) | 0.849 | | |
| Frailty index | FN-BMD | 14 | IVW | 0.034 (- 0.170, 0.237) | 0.746 | 8.917 | 0.779 |
| | | | WM | 0.049 (- 0.227, 0.324) | 0.729 | | |
| | | | MR Egger | - 0.106 (- 1.256, 1.044) | 0.860 | | |
| Frailty index | LS-BMD | 14 | IVW | 0.048 (- 0.223, 0.320) | 0.727 | 17.140 | 0.193 |
| | | | WM | 0.101 (- 0.217, 0.418) | 0.534 | | |
| | | | MR Egger | - 0.044 (- 1.631, 1.544) | 0.958 | | |

Abbreviations: MR, Mendelian randomization; BMD: bone mineral density; FN: femoral neck; LS: lumbar spine; FA: forearm
e: heel; SNPs, single nucleotide polymorphisms; IVW, inverse-variance weighted; CI, confidence interval; WM, weighted median.

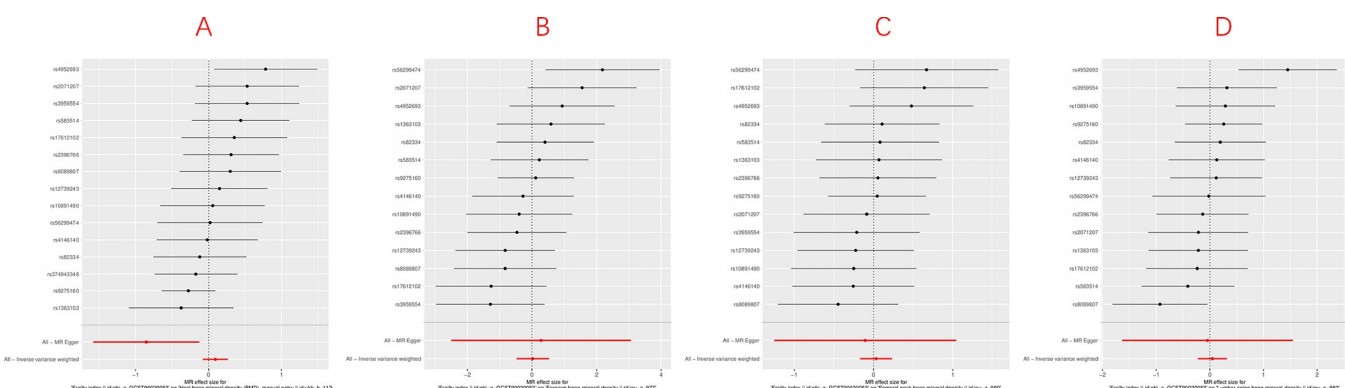

**Fig 3.** Forest plot of the causal effects of frailty index (FI) associated SNPs on bone mineral density (BMD) Figure showed the Mendelian randomization estimated effects sizes for (A) heel (e) BMD, (B) forearm (FA) BMD, (C) femoral neck (FN) BMD, (D) lumbar spine (LS) BMD. Data are expressed as $\beta$ values with 95% CI.

0.320, $P$ = 0.727). In addition, the combined results of the MR-Egger intercept test and the MR-PRESSO global test showed that the MR analysis was not affected by any potential horizontal pleiotropy ($P > 0.05$) (**Table 3**).

Scatter plots and funnel plots are presented in **S1 File**. Funnel plots were used to assess potential asymmetry, which could indicate the presence of directional horizontal pleiotropy. To further investigate potential pleiotropic effects, we conducted MR-Egger intercept test and MR-PRESSO global test, which are robust to directional pleiotropy. These analyses help validate the robustness and reliability of our MR results, providing evidence that our causal effect estimates were not influenced by individual SNP or substantial pleiotropy bias.

## Discussion

In this study, we used summary statistics from the GWAS to identify the causal relationship between BMD and frailty. This bidirectional MR analysis suggests potential evidence for a negative causal relationship between both e-BMD and FA-BMD with FI, indicating that lower BMD at these skeletal sites may contribute to increased frailty. In contrast, the FI has no causal effect on BMD at different skeletal sites. It is worth noting that our study is the first MR investigation to examine the bidirectional causal relationship between BMD at different skeletal sites

**Table 3. Results of horizontal pleiotropy by the MR-Egger intercept test and MR-PRESSO global test.**

| Exposure(s) | Outcome(s) | MR-Egger intercept test | | MR-PRESSO global test | |
|---|---|---|---|---|---|
| | | Intercept | *P*-value | RSS obs | *P*-value |
| e-BMD | Frailty index | <0.001 | 0.705 | 66.778 | 0.094 |
| FA-BMD | Frailty index | 0.002 | 0.798 | NA | NA |
| FN-BMD | Frailty index | 0.004 | 0.347 | 24.755 | 0.324 |
| LS-BMD | Frailty index | 0.001 | 0.733 | 37.000 | 0.051 |
| Frailty index | e-BMD | 0.022 | 0.021 | 19.383 | 0.291 |
| Frailty index | FA-BMD | - 0.005 | 0.860 | 21.055 | 0.165 |
| Frailty index | FN-BMD | 0.003 | 0.813 | 10.183 | 0.803 |
| Frailty index | LS-BMD | 0.002 | 0.910 | 19.819 | 0.207 |

Abbreviations: MR, Mendelian randomization; BMD: bone mineral density; FN: femoral neck; LS: lumbar spine; FA: forearm
e: heel; RSS, residual sum of squares.

and frailty. This contributes to our understanding of the relationship between bone health and frailty in the context of genetic associations. Total body BMD was not explored in our study, as it is usually measured in children [6].

Several previous observational studies have shown that low BMD is associated with an increased risk of frailty. S.-L. Ma et al. found that self-reported frailty was associated with low e-BMD during the 6-year follow-up in 230 elderly (82% female, 58% African–American) [33]. Our study further enhanced the evidence supporting the causal impact of low e-BMD on the progression of frailty. This was accomplished by employing a MR approach, which encompassed a larger sample size and a gender-balanced composition, thereby improving the validity of our findings. Anne M. Kenny et al. evaluated 392 community-dwelling men (age range: 58–95 years) and found that increased frailty was associated with lower FN-BMD, although the association was not independent of age [34]. MICHAEL J. COOK et al. conducted a cross-sectional survey of 3231 European male subjects, the result indicated frail men had lower FN-BMD compared to robust men ($P < 0.05$), but not lower LS-BMD [8]. In a survey of 1839 participants in Taiwan, Li-Kuo Liu et al. found that frailty was significantly associated with lower hip BMD [9]. Our findings of no causal effects of FN-BMD and LS-BMD on FI are inconsistent with the results of the observational studies mentioned above. It is important to note that degenerative changes in the spinal region among the elderly can artificially elevate BMD readings, thereby introducing potential inaccuracies in observational studies and subsequently affecting the reliability of the results. Risk factors for frailty encompass a broad spectrum of aspects and conditions, spanning sociodemographic, clinical, lifestyle-related, and biological domains [35]. In the elderly population, sarcopenia and osteoporosis frequently coexist and exhibit a close association with frailty [36]. Some studies showed that muscle defects accounted for a greater proportion of the risk for frailty than bone defects [10, 37]. Despite the available evidence highlighting various risk factors for frailty, our understanding of the underlying biological mechanisms leading to its development remains limited [2].

Likewise, the causal effect of frailty on site-specific BMD remains uncertain. A prospective study of 235 community-dwelling women by S.A. Sternberg et al. showed that frailty defined by the Vulnerable Elders Survey could predict a decrease in BMD after 1 year [38]. In a systematic review conducted by Kojima, six studies involving a total of 96,564 community-dwelling older individuals were analyzed. The findings revealed that individuals who were classified as frail or pre-frail exhibited a significantly higher likelihood of experiencing fractures [39]. However, using three different MR estimation methods (IVW, WM, and MR-Egger), we did not find a causal relationship between FI and BMD (different skeletal sites). It is possible that different frailty phenotypes or scoring criteria were used and that there was subjectivity in the determination of frailty. Indeed, observational studies often suffer from confounding factors or reverse causality, which can be avoided in MR studies. We use the Frailty Index (FI), which is based on the accumulation of many health deficits over the course of life. The underlying mechanisms of FI are multifactorial and currently unknown, but a genetic basis has been proposed, with heritability estimates ranging from 30% to 45% [40, 41].

This study has several limitations that should be considered when interpreting the results. First, The observed potential negative causality between the e-BMD dataset and the FI dataset must be interpreted with caution due to a substantial sample overlap, which could increase the risk of false positive findings. This limitation must be carefully considered when evaluating the results of our analysis. As the availability of GWAS datasets expands, we anticipate conducting future research without such overlap to validate and refine our findings. Second, our study is potentially limited by the relatively small number of SNPs that were used as genetic instruments. The power of a Mendelian randomization analysis to detect causal relationships is partly contingent on the proportion of genetic variance in the exposure accounted for by the

instruments. With a limited set of SNPs, our analysis may not have captured enough of the genetic variability associated with our exposure to conclusively determine causality. Future studies with larger GWAS datasets could employ a more extensive array of SNPs, enhancing the power and reliability of MR findings. The third is horizontal pleiotropy. While using a large number of genetic variants associated with BMD as strong instruments is valuable, it is crucial to acknowledge that these variants may not be thoroughly characterized. As a result, it is possible that pleiotropic variants that affect pathways distinct from BMD could have been introduced. This highlights the importance of carefully examining potential pleiotropic effects and conducting sensitivity analyses to ensure the robustness of the findings. Our sensitivity analysis combining multiple pleiotropic robust MR methods provided essentially similar results. Fourth, our analysis included participants of predominantly European ancestry, thus limiting the generalization of findings to other ethnic populations. To ensure that the insights gleaned from Mendelian randomization are universally relevant, further studies incorporating diverse ethnic groups are essential. Such research would provide a more comprehensive understanding of the genetic underpinnings associated with the phenotypes of interest and enable a broader application of the findings.

Despite the limitations, the present study also boasts several advantages that reinforce the validity of our findings. First, we utilized a comprehensive approach by including genetic variants associated with BMD not only from a single skeletal site but from multiple anatomical locations, capturing a more systemic genetic influence on BMD. This broad-based selection of BMD-associated variants may enhance our understanding of the relationship between bone density and health outcomes across different parts of the skeleton. Second, the utilization of MR methods brings further advantages by enabling the extraction of a large amount of genetic data from publicly available datasets. This approach not only saves time and resources but also enhances statistical power, which is especially important in genomic studies. Finally, our choice of IVs for the FI was derived from the largest GWAS available for this trait. The use of such a robust GWAS dataset increases the likelihood that the IVs selected are truly representative of the genetic determinants of frailty, potentially leading to more precise and reliable Mendelian randomization estimates.

## Conclusion

In conclusion, the findings of this MR study offer suggestive evidence that low bone mineral density (BMD) may be a potential risk factor for frailty, although the support for this assertion is not definitive. However, the evidence did not support a causal relationship between frailty and BMD in the inverse analysis. These findings present an opportunity for further exploration and understanding of the associations between frailty and osteoporosis. Such understanding can be valuable in the development of preventive strategies for both frailty and osteoporosis.

## Supporting information

**S1 File. Supplementary table, figures, and R code are included in the file.**
(DOCX)

**S2 File. SNPs used as instrumental variables are included in the file.**
(XLSX)

## Acknowledgments

The authors thank all the investigators and participants for sharing the GWAS data.

## Author Contributions

**Conceptualization:** Jue-xin Shen.

**Data curation:** Yi Lu.

**Writing – original draft:** Yi Lu.

**Writing – review & editing:** Jue-xin Shen, Wei Meng, Lei Yu, Jun-kai Wang.

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
