## [Decision Letter · Decision Letter 0]

2 Nov 2023

PONE-D-23-23648Exploring causality between bone mineral density and frailty: a bidirectional Mendelian randomization studyPLOS ONE

Dear Dr. Lu,

Thank you for submitting your manuscript to PLOS ONE. After careful consideration, we feel that it has merit but does not fully meet PLOS ONE’s publication criteria as it currently stands. Therefore, we invite you to submit a revised version of the manuscript that addresses the points raised during the review process.

The manuscript is necessary to be major-revisioned according to the Reviewer's comments.

We look forward to receiving your revised manuscript.

Kind regards,

Masaki Mogi

Academic Editor

PLOS ONE

https://doi.org/10.1016/j.joca.2021.12.001

In your revision ensure you cite all your sources (including your own works), and quote or rephrase any duplicated text outside the methods section. Further consideration is dependent on these concerns being addressed.

Reviewers' comments:

Reviewer's Responses to Questions

**Comments to the Author**

1. Is the manuscript technically sound, and do the data support the conclusions?

Reviewer #1: Yes

2. Has the statistical analysis been performed appropriately and rigorously? 

Reviewer #1: No

3. Have the authors made all data underlying the findings in their manuscript fully available?

Reviewer #1: Yes

4. Is the manuscript presented in an intelligible fashion and written in standard English?

Reviewer #1: Yes

5. Review Comments to the Author

Reviewer #1: In the manuscript “Exploring causality between bone mineral density and frailty: a bidirectional Mendelian randomization” by Jue-xin Shen, Yi Lu, Wei Meng, Lei Yu and Jun-kai Wang, the authors perform a bidirectional Mendelian randomization analysis of the relationship between bone mineral density and frailty. It is an interesting topic, however, the manuscript needs some revisions.

A great thing about Mendelian randomization which uses publicly available GWAS SS data is that it is fully reproducible, so I have performed all of the analyses stated in the manuscript on my own computer and was able to check the results.

Here are my comments:

Abstract

Line 29: “We performed genome-wide association statistics for BMD” – please correct this, as you did not perform a GWAS.

Line 32: “different skeletal sites (N=195,723)” – please state the exact sample size for each of the skeletal sites analysed.

Line 34: “We found that low heel (e) BMD led to an increase in FI” – please rephrase this to say that you found a negative causal estimate between genetically predicted heel BMD and FI. Your study is not a longitudinal study and the use of phrases such as ‘led to increase’ should be avoided.

Line 34-37: Report either the confidence interval or the p-value, as these two give the same information.

Introduction

Line 55: Please give a reference for his statement.

Methods

Line 107: Please replace ‘pooled data’ with ‘summary statistics’.

Line 122-126: Please state the sample size for each of the sites analysed in the manuscript text (as you did in the S1 File) as the 53236 samples does not refer to all of them.

Line 127-128: This is where you have a problem. “e-BMD dataset were from the UKB GWAS summary statistics, including 142,487 participants of European descent” – you have a substantial overlap between your FI dataset (which is based on a GWAS in UK Biobank) and the heel BMD which is also based on a GWAS in UKB. This leads to biased causal estimates because of sample overlap. Ideally, the exposure and outcome datasets in a two-sample MR setting should not be overlapping. For a continuous outcome, bias due to sample overlap is a linear function of the proportion of overlap between the samples (see https://doi.org/10.1002/gepi.21998). I have checked the GWAS catalog and the ieu open gwas project. There is only one GWAS on frailty index and all of the GWASs on heel BMD are based on UKB. So, I am aware that you will not be able to find a dataset that does not overlap (correct me if I’m wrong), meaning that you need to be aware that your only significant finding in this manuscript could be a false positive.

Line 133-135: “First, we utilized the PLINK clumping method within the two-sample MR tool to aggregate SNPs that were significantly associated with the exposure in the GWAS summary data.” – this is not what you did as I can see from the code that you are not using PLINK but used the built-in function extract_instruments from the TwoSampleMR package which returns a set of LD clumped SNPs that are GWAS significant for your exposure. Also, keep in mind that clumping is used for pruning based on LD (second step), and that the first step is just setting a significance threshold to find the significantly associated SNPs.

Line 136-140: “To ensure the independence of the selected instrumental variables, we further excluded SNPs that showed linkage disequilibrium (LD) based on European data from the 1,000 Genomes Project as a reference panel (R2 > 0.001 and kb < 10,000) [21].” – again, this was all done by using the built-in extract_instruments function. Please reference the function and the package.

Line 140-141: “ This step helps ensure that the instrumental variables used in the analysis are not correlated due to their proximity to the genome.” – what is a proximity to the genome?

Line 159: change ‘signed’ to ‘assigned’.

Line 163-164: “To detect and correct for pleiotropy, we also performed two analyses” – the MR-Egger intercept is a trivial method to check for horizontal pleiotropy, I just don’t see how can you check for horizontal pleiotropy by using weighted median? Also, neither of these two can correct for horizontal pleiotropy. If you want to correct for uncorrelated horizontal pleiotropy then you can use MR-PRESSO for example, and if you want to correct for correlated horizontal pleiotropy you can use the CAUSE method.

Line 178-179: Specify the “problems with multiple testing”.

Line 182: Thank you for providing the R code.

Results

Line 192: Where do these 557 SNPs come from? When I extract the instruments for each of the BMD measure sites separately, I get 513 instruments for heel BMD (ebi-a-GCST006979), 21 instruments for femoral neck BMD (ieu-a-980), 3 instruments for forearm BMD ('ieu-a-977') and 24 instruments for lumbar spine BMD (ieu-a-982).

You should divide the results section by each of the exposure analysed: 1. Heel BMD, 2. FN BMD, 3. FA BMD and 4. LS BMD, so 4 paragraphs in the Results section for the forward direction MR.

Also, you have an error in your code: exp_dat <- extract_instruments(c('ebi-a-GCST006979','ieu-a-980','ieu-a-977','ieu-a-982'))

You are extracting all of the exposures together, this then gives you a combined dataset of 561 SNPs from all 4 exposures and in the next step you are using the: out_dat <- extract_outcome_data(snps = exp_dat$SNP, outcomes = 'ebi-a-GCST90020053'), which is extracting SNPs significant and LD-pruned for all 4 exposures together from the outcome dataset. This is incorrect and you should separate the analyses for each of the exposure-outcome combinations, for example:

exp_dat <- extract_instruments('ebi-a-GCST006979')

and then

out_dat <- extract_outcome_data(snps = exp_dat$SNP,

outcomes = 'ebi-a-GCST90020053')

Line 193-194: “All of them are associated with FI at genome-wide significance.” – this is not true.

Line 204-205: “Lastly, 543 SNPs were taken as IVs for BMD” – BMD is not one exposure, but 4, please correct this.

The same remarks apply to the presentation of results in the reverse direction. Please, divide it by the 4 outcomes, so 4 paragraphs in the Results section for the reverse direction MR.

Also, there is no need to perform the leave-one-out analysis if the primary finding is not significant.

Supplementary files: Try to combine them into fewer files. For example, file S1 is just 1 table, it can be combined with file S2 and file S5.

Both files S3 and S4 should be formatted in a way that is easy to read, right now it is impossible to read them as they are in the csv format. Use the library(WriteXLS) to write these data into an xlsx file and combine the two files into one xlsx file with 2 sheets.

6. PLOS authors have the option to publish the peer review history of their article (what does this mean?). If published, this will include your full peer review and any attached files.

Reviewer #1: **Yes: **Nikolina Pleić

---

## [Author Response · Author response to Decision Letter 0]

5 Dec 2023

Dear Academic Editors and Reviewers,

We would like to express our sincerest thanks for the valuable insights and detailed feedback provided on our manuscript. Your constructive comments have been instrumental in enhancing the overall quality and clarity of our work.

In the attached "Response to Reviewers" document, we have organized the comments from academic editors and reviewers for clear reference. Each comment is presented in italics, and to facilitate a thorough review, they are numbered. Our responses are provided in plain text, and any modifications to the manuscript text are indicated by highlighting in yellow.

We believe that we have addressed all the points raised and that the revisions have substantially improved our manuscript. We are pleased to resubmit our revised manuscript for your consideration, and we are hopeful that it now meets the high standards of [PLOS ONE].

Thank you once again for your time and the opportunity to refine our manuscript with your guidance.

Yours sincerely,

Yi Lu

---

## [Decision Letter · Decision Letter 1]

26 Dec 2023

Exploring causality between bone mineral density and frailty: a bidirectional Mendelian randomization study

PONE-D-23-23648R1

Dear Dr. Lu,

We’re pleased to inform you that your manuscript has been judged scientifically suitable for publication and will be formally accepted for publication once it meets all outstanding technical requirements.

Kind regards,

Masaki Mogi

Academic Editor

PLOS ONE

Additional Editor Comments (optional):

Reviewers' comments:

Reviewer's Responses to Questions

**Comments to the Author**

1. If the authors have adequately addressed your comments raised in a previous round of review and you feel that this manuscript is now acceptable for publication, you may indicate that here to bypass the “Comments to the Author” section, enter your conflict of interest statement in the “Confidential to Editor” section, and submit your "Accept" recommendation.

Reviewer #1: All comments have been addressed

2. Is the manuscript technically sound, and do the data support the conclusions?

Reviewer #1: Yes

3. Has the statistical analysis been performed appropriately and rigorously? 

Reviewer #1: Yes

4. Have the authors made all data underlying the findings in their manuscript fully available?

Reviewer #1: Yes

5. Is the manuscript presented in an intelligible fashion and written in standard English?

Reviewer #1: Yes

6. Review Comments to the Author

Reviewer #1: (No Response)

7. PLOS authors have the option to publish the peer review history of their article (what does this mean?). If published, this will include your full peer review and any attached files.

Reviewer #1: **Yes: **Nikolina Pleić

---

## [Editor Report · Acceptance letter]

10 Jan 2024

PONE-D-23-23648R1 

PLOS ONE

Dear Dr. Lu, 

I'm pleased to inform you that your manuscript has been deemed suitable for publication in PLOS ONE. Congratulations! Your manuscript is now being handed over to our production team.

Kind regards, 

on behalf of

Dr. Masaki Mogi 

Academic Editor

PLOS ONE